# Hydrocolloid Coatings as a Pre-Frying Treatment in the Production of Low-Fat Banana Chips

Júlia Silva Pereira dos Santos [1], Magali Leonel [1,*], Paulo Ricardo Rodrigues de Jesus [2], Sarita Leonel [1,2], Adalton Mazetti Fernandes [1] and Lucas Felipe dos Ouros [1]

[1] Center for Tropical Roots and Starches (CERAT), São Paulo State University (UNESP), Botucatu 18610-034, SP, Brazil; juhsantosps@gmail.com (J.S.P.d.S.); sarita.leonel@unesp.br (S.L.); adalton.fernandes@unesp.br (A.M.F.); lucas.ouros@unesp.br (L.F.d.O.)

[2] School of Agriculture (FCA), São Paulo State University (UNESP), Botucatu 18610-307, SP, Brazil; pr.jesus@unesp.br

[*] Correspondence: magali.leonel@unesp.br; Tel.: +55-14-38807612

**Abstract:** Fried foods occupy a large portion of the fast-food market. However, growing consumer health awareness is driving research to minimize the oil content of products. The use of specific barriers such as hydrocolloid coatings can avoid high oil absorption. Herein, the physicochemical characteristics of banana fruit cultivars and the effects of hydrocolloid coatings on the quality attributes of banana chips were evaluated. The unripe fruits were analyzed for length, diameter, mass, pulp/peel ratio, firmness, and color. The pulps were analyzed for moisture, total and reducing sugars, crude fat, total soluble solids, pH, titratable acidity, and maturation index. Pulp slices were coated before frying with four hydrocolloid solutions: guar gum, carboxymethylcellulose, sodium alginate, and maltodextrin. The fruits of the cultivar BRS Platina were larger, with a more yellowish pulp and those of FHIA 18 had smaller lengths and diameters. After frying, higher moisture losses were observed in the control and in the treatment with maltodextrin. Hydrocolloids promoted reductions in oil absorption from 26.54 to 9.14%, in addition to changes in the color and hardness of the chips. A coating pretreatment can be used to produce low-fat banana chips, with better results using guar gum. Our results are useful in expanding the commercialization of fruits that fall easily and the chip coating technology can be applied in snack industries.

**Keywords:** edible coating; fried products; *Musa* spp.; oil absorption

## 1. Introduction

Bananas are predominantly produced in Asia, Latin America, and Africa. Global banana production in 2021 was 124.9 million tons, with India being the world's largest producer (33.06 million tons) and Brazil ranking fourth with 6.81 million tons [1]. According to the Food and Agriculture Organization of the United Nations, food waste and food loss are considered a major issue. Ensuring responsible consumption and production patterns is among the Sustainable Development Goals (SDGs), with the forecast that, by 2030, per capita food waste and losses in the world will be reduced by 50% throughout the production and supply chains, including post-harvest losses [1–4].

The banana fruit is a healthy, nutritious fruit, which contains 75% moisture, 23% carbohydrates, 1% protein, and 0.5% fat. The banana crop, despite its high worldwide production, suffers losses of up to 40% of the total global production, due to conditions such as the perishability of the fruit and the low technology used by producers [5,6].

Growing from the north to the south of Brazil, bananas are a cheap source of energy, minerals, and vitamins. However, the country, like other world producers, has high post-harvest waste as one of the main challenges to be minimized [7]. The main banana cultivars in Brazil are Prata, Grand Naine, Williams, and Nanicão. However, the incentive to plant other cultivars has been constant, given the importance of germplasm diversification for the

advancement of banana growing. Another strategy has been to encourage the processing of cultivars that have higher agronomic performance but post-harvest disadvantages, such as easy dropping and fruit with a taste that is not appreciated by consumers [8,9].

The production of fried banana chips has been an alternative to minimize production losses and introduce cultivars suitable for industrial processing [10–13]. The global fried snack market is constantly growing having been valued at USD 536.5 billion in 2022, growing at a CAGR of 5.61% from 2022 to 2023 [14]. However, consumers' increased health awareness has affected their purchasing choices and has directed studies towards production technologies that reduce the oil content in fried snacks [15–18].

Frying is a thermal process involving heat and mass transfer. During the process, chemical changes occur in the product, such as starch gelatinization, protein denaturation, surface darkening, water evaporation, and oil absorption. The frying process by immersion in oil takes place at high temperatures, causing the absorption of a large amount of fat. Oil absorption is affected by several factors such as frying temperature and time, food composition, exposure area, pre-treatments, and oil quality [19,20].

Hydrocolloids are long-chain, high-molecular-weight hydrophilic polymers with various functional groups, such as amino and carbonyl groups. These polymers can be classified according to their sources, chemical structures, and ionic properties. They are readily dispersive, fully or partially soluble, and are prone to swelling in water, changing the physical properties of the solution to form gels or allow for thickening, emulsification, coating, and stabilization [21,22].

The ability of hydrocolloids to form films and coatings gives them the potential to reduce oil absorption in fried products. Furthermore, the use of hydrocolloids can prevent the formation of acrylamide, a processing-induced contaminant, with a positive correlation between reducing sugar content and acrylamide formation, especially with increasing fruit maturity [23,24]. Hydrocolloids commonly used as edible films and coatings include gums such as guar and xanthan, cellulose derivatives such as methylcellulose, hydroxypropylcellulose, and carboxymethylcellulose, and other polysaccharides and proteins [19].

Considering the importance of reducing losses in banana growing, as well as the need to produce healthy products, the aim of this work was firstly to analyze the physicochemical characteristics of the fruits of two banana cultivars, with a view to using them as raw materials for the production of fried chips. Subsequently, the study aimed to evaluate the use of four types of hydrocolloids as a pre-treatment for the frying process, aiming to produce low-fat banana chips. The data generated by this study are of great interest to banana growers and the industry, who are looking to expand the banana market by obtaining attractive and healthier products for consumers.

## 2. Materials and Methods

### 2.1. Orchard and Fruit Harvest

The banana orchard is located at the School of Agriculture, São Paulo State University (FCA/UNESP), Botucatu city, São Paulo State, Brazil. Botucatu city is located at 22°51′55″ S, 48°27′22″ W and at an altitude of 810 m (a.s.l.). The prevailing climate is temperate mesothermal (Cfa) according to the Köppen classification, and with an average temperature close to 22 °C [25]. The soil in the experimental area is Red Nitosol [26]. The orchard has been in production since 2017, and the management and cultural practices follow the recommendations for cultivation in the region.

Cultivar BRS Platina, also known as PA42-44, is a traitploid hybrid (AAAB) derived from crossing the cultivar Prata-Anã (AAB) with the diploid M53 (AA) [27]. It shows resistance to Yellow Sigatoka (*Mycosphaerella musicola*) and Panama disease (*Fusarium oxysporum* f.sp. *cubense*), good tillering, a cycle of approximately 375 days, and an average productivity of 20 t ha$^{-1}$. The "BRS Platina" is a viable alternative to increase the number of Prata varieties available to the growers, especially in areas free of Black Sigatoka, such as the Northeast and part of Southeast Brazil, and where there is the presence of Panama disease [27,28]. The cultivar FHIA 18 was developed by the Honduran Agricultural Re-

search Foundation (FHIA) breeding program. FHIA 18 is a synthetic tetraploid hybrid (AAAB) originating from the plants of a Prata Ana (AAB, Pome) × SH-3142 cross. The fruits of FHIA 18 are of the Prata type and fall with ease. The plants are desired for their high yield (bunches up to 40 kg and 10 hands), the ability to grow at lower temperatures, and an overall cold tolerance. The "FHIA 18" banana tree has great potential to replace the Prata cultivar in several regions of Brazil [29–31]. These cultivars, despite having good agronomic performance and resistance to the main diseases, require incentives for processing, as both are highly susceptible to fruit fall, a characteristic that affects handling, transport, and marketing, reducing the commercial value and acceptance of the fruits by consumers [29–31].

Ten plants per cultivar were randomly selected and bunches with green fruit were harvested in the morning.

### 2.2. Characteristics of Unripe Fruits of Banana Cultivars

The physical analyses carried out on unripe banana fruits measured the fruit length, fruit diameter, fruit mass (whole, pulp, peel, pulp/peel ratio), peel and pulp color, and firmness of the unpeeled fruit and pulp.

The longitudinal length was measured with a tape measure on the outside of the fruit and the results are expressed in centimeters. The diameter was measured with a caliper and the results are expressed in millimeters. Measurements were taken at three different points on the fruit (close to the peduncle, diameter in the middle of the fruit, and diameter close to the apex). The fruits were weighed on a semi-analytical digital scale (BEL, model RB 16001, Piracicaba, SP, Brazil) and the masses of the whole fruit, peel, pulp, and pulp/peel ratio were determined.

Fruit color was determined by direct reading using a digital colorimeter (CR 400, Konica Minolta, Ramsey, NJ, USA). The illuminant $D_{65}$ was used. The color was expressed as the average of three L*, a*, and b* readings, where L* stands for brightness, +a* redness, −a* greenness, +b* yellowness, and −b* blueness [32]. Chromaticity and hue angle were calculated. A white calibration plate was used to standardize the equipment prior to color measurements (Y = 93.69, x = 0.3170, y = 0.3335). The color of the banana peel is not uniform across the entire surface; therefore, measurements were made in the three regions of the fruit (close to the peduncle, middle of the fruit, and close to the apex) and the average values were considered.

Firmness analysis was performed on a texture analyzer (TA. XT plus stable micro-systems, Surrey, UK). The measurement was performed using a TA-52 (2 mm) probe with the following operating conditions: penetration distance into the sample of 20 mm, pre-test speed of 1.0 mm/s, test speed into the sample of 2.0 mm/s, and post-test speed of10 mm/s. The results are expressed as the mean of three measurements.

Fruit pulps were analyzed for pH, titratable acidity, soluble solids, ratios, moisture, total sugars, reducing sugars, and crude fat. The soluble solid content was evaluated with a portable digital refractometer, with readings in the range of 0 to 32 °Brix. The titratable acidity was evaluated by titration to pH 8.2 under constant stirring with a NaOH solution (0.05 N). The results are expressed in grams of malic acid per 100 g of pulp. The maturation index (ratio) was calculated using the SS/TA ratio (soluble solids/titratable acidity). The AOAC methods were followed for the determination of moisture (934.06), total sugars (968.28), reducing sugars (945.66), and crude fat (945.16) [33].

### 2.3. Production of Banana Chips

The process was defined based on previous studies [10–14,16,19] and preliminary experiments. The bananas were peeled and cut into 2 mm thick slices. The sliced bananas were immediately immersed in a 0.3% (*w/v*) citric acid solution for 10 min and drained to inhibit the enzymatic browning of raw banana slices.

Four hydrocolloid solutions (carboxymethylcellulose, guar gum, sodium alginate, and maltodextrin) at a level of 1.5% (*w/w*) were prepared. Each gum powder was separately

dispersed and dissolved in distilled water at 75 °C with continuous and slow stirring to obtain a uniform solution. The resulting solutions were then cooled to room temperature. Then, the banana slices were immersed in the solutions for 2 min in a 15:1 ratio of banana to solution. After coating, the samples were drained, dehydrated in an air circulation oven (135 °C/3 min) to reduce surface moisture, and cooled to room temperature. A stainless-steel sieve was used to collect the coated slices and they were allowed to drain for 30 s. The weight of the coating solution before and after dipping the slices was recorded to determine the weight gain.

Coated and uncoated banana slices were fried. Frying was carried out in a fryer with thermostatically controlled temperature (Scavone, SP, Brazil). First, the fryer was filled with 6 L of sunflower oil and the oil was preheated. All banana slices were fried at 170 °C for 4 min with a product weight/oil volume ratio of 1:32. At the end of frying, the basket was lifted, and the samples were allowed to drain for 2 min. The oil level volume and temperature were checked and confirmed after each frying. Five batches of 200 g of banana samples were fried per treatment and at the end of each treatment, the oil was replaced.

Figure 1 summarizes the steps followed in this study.

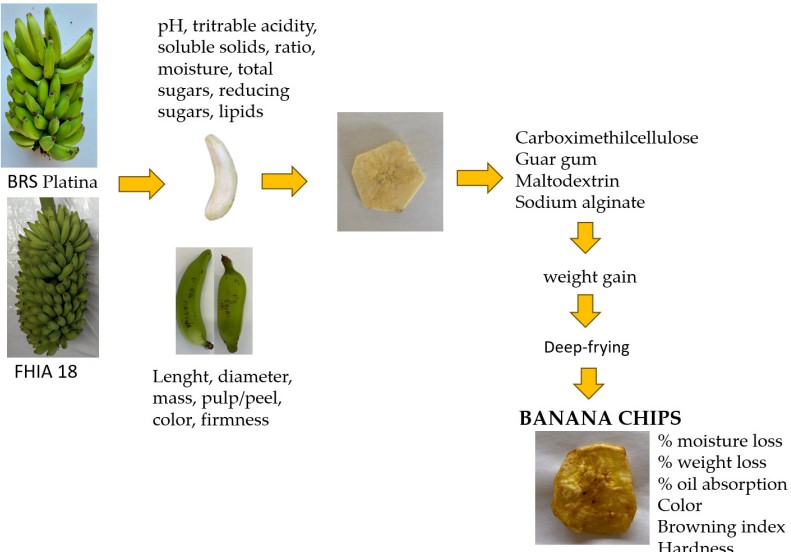

**Figure 1.** Flowchart of banana chip production.

### 2.4. Characteristics of Banana Chips

Fried banana chips were analyzed for moisture, lipids, color, and texture. The moisture content of the chips was determined according to method 934.06 [33]. Moisture loss was calculated as the ratio between the moisture content of raw slices before treatment and that of fried slices.

$$\% moisture\,loss = \frac{moisture\,of\,raw\,slices - moisture\,of\,fried\,slices}{moisture\,of\,raw\,slices} \times 100$$

The oil content was determined according to method 945.16 [33] and the percentage of reduction in oil absorption was calculated as the ratio between the lipids of the control treatment chips and that obtained in the chips submitted to the pretreatments with hydrocolloids.

$$\% reduction\,of\,oil\,absorption = \frac{lipids\,of\,control\,treatment - lipids\,of\,coated\,chips}{lipids\,of\,control} \times 100$$

Fried banana chip color was measured with a Minolta colorimeter (CR 400, Konica Minolta, NJ, USA) using the CIELAB color parameters L*, a*, and b* [33]. The hue angle (h°) and browning index (BI) were calculated.

$$\Delta E = \sqrt{\left(\Delta L^*\right)^2 + \left(\Delta a^*\right)^2 + \left(\Delta b^*\right)^2}$$

$$BI = \frac{100 \times (X - 0.31)}{0.17}$$

where

$$X = \frac{a^* + \left(1.75 \times L^*\right)}{\left(5.645 \times L^*\right) + a^* - \left(3.012 \times b^*\right)}$$

The hardness of the samples was measured using a texture analyzer (TA.XT plus stable micro-systems, Surrey, UK). The hardness of the fried banana chips was defined as the maximum compressive force during the first bite and determined from the force–strain curves. A crisp chip/cracker rig + 1/4″ rounded end probe was used. Test settings were a pre-test velocity of 1.00 mm/s, test velocity of 2.00 mm/s, post-test velocity of 10.00 mm/s, and trigger force of 20.0 g. A low value of hardness indicates high crispiness.

### 2.5. Statistical Analysis

A completely randomized experimental design was used to study the effect of cultivars and type of hydrocolloid. Five replicates were used to determine each property. The software package SISVAR, version 5.6 (Lavras, Brazil) was utilized to perform analysis of variance (ANOVA). Tukey's test was used to determine the significant differences between treatments ($p < 0.05$).

## 3. Results and Discussion

### 3.1. Characteristics of Unripe Fruits of Banana Cultivars

Bananas are available in different sizes, shapes, and colors. When selecting banana cultivars for frying, the physical characteristics of the fruits (size, color, texture, pulp/peel ratio) are important for the yield of the process. These characteristics together with the chemical composition of the pulp indicate the degree of maturity of the fruit, which will have a direct impact on the quality of the final product.

The results of the analysis of physical characteristics of the banana fruits are shown in Table 1. According to the Brazilian Program for the Modernization of Horticulture [34], bananas are classified by fruit length into five classes, with class 12 containing fruits with a length between 12 and 15 cm, the length observed for the fruits of FHIA 18. The cultivar BRS Platina has fruits in class 15 (length > 15 and <18). The fruit category is determined by caliber (diameter of the central part of the fruit). The fruits of BRS Platina were classified as category extra (34 mm) and those of FHIA 18 in category II (28 mm). Leonel et al. [9] observed that the BRS Platina fruits grown in the same orchard as this study harvested in 2017 and 2018 had a length ranging from 16.73 to 17.75 cm, diameter from 36.66 to 37.05 mm, and mass from 126.33 to 127.49 g.

Variations in fruit dimensions were reported by Aquino et al. [35], who observed a total length of 13.02 to 24.58 cm, and a central diameter of 29.82 to 49.96 mm for 15 banana cultivars. According to these authors, the AA cultivars have small and cylindrical fruits, and the peel is thin and more adhered to the pulp, unlike the triploid cultivars and tetraploids, which have larger fruits, with a firm pulp, marked edges, and thicker peel, which are interfering factors for the physical measurements of the fruits.

In a study evaluating banana cultivars for the frying industry, a fruit length greater than 14 cm and pulp diameter greater than 3.5 cm are desired quality factors, so the BRS Platina cultivar meets these requirements [12]. The fruits of the FHIA 18 cultivar are smaller

in length and diameter, which reduces the yield in slices, but allows the production of chips with a smaller diameter, which can be an interesting factor for packaging small portions.

**Table 1.** Physical characteristics of unripe banana fruits.

| Parameter | Cultivars | | CV (%) | MSD |
|---|---|---|---|---|
| | **BRS Platina** | **FHIA 18** | | |
| Fruit length (cm) | 16.76 ± 0.75 a | 13.50 ± 0.43 b | 8.73 | 2.78 |
| Fruit diameter (peduncle) (mm) | 39.18 ± 1.49 a | 23.50 ± 1.12 b | 3.04 | 2.14 |
| Fruit diameter (middle of the fruit) (mm) | 40.57 ± 1.27 a | 30.01 ± 1.41 b | 5.92 | 4.70 |
| Fruit diameter (apex) (mm) | 36.15 ± 0.97 a | 27.00 ± 1.22 b | 6.74 | 4.79 |
| Fruit mass (g) | 131.00 ± 2.37 a | 59.73 ± 1.44 b | 1.98 | 4.25 |
| Peel mass (g) | 42.42 ± 0.40 a | 23.63 ± 0.71 b | 1.73 | 1.29 |
| Pulp mass (g) | 74.12 ± 1.71 a | 30.65 ± 0.72 b | 5.50 | 6.48 |
| Pulp/peel | 1.75 ± 0.06 a | 1.32 ± 0.04 b | 3.63 | 0.28 |
| Firmness whole fruit (peduncle) (N) | 109.37 ± 0.39 a | 89.95 ± 0.74 b | 6.46 | 1.45 |
| Firmness whole fruit (middle of the fruit) (N) | 112.50 ± 0.62 a | 88.70 ± 0.73 b | 9.61 | 2.17 |
| Firmness whole fruit (apex) (N) | 110.92 ± 0.42 a | 88.85 ± 0.72 b | 7.48 | 1.69 |
| Peel firmness (peduncle) (N) | 70.37 ± 0.38 a | 51.97 ± 0.56 b | 9.43 | 1.38 |
| Peel firmness (middle of the fruit) (N) | 77.70 ± 0.43 a | 57.00 ± 0.65 b | 4.14 | 0.63 |
| Peel firmness (apex) (N) | 77.50 ± 0.48 a | 54.45 ± 0.61 b | 2.21 | 0.33 |
| Pulp firmness (peduncle) (N) | 60.61 ± 0.49 a | 60.12 ± 0.54 a | 8.70 | 1.18 |
| Pulp firmness (middle of the fruit) (N) | 59.70 ± 0.39 a | 57.10 ± 0.26 b | 4.35 | 1.88 |
| Pulp firmness (apex) (N) | 60.05 ± 0.35 a | 58.52 ± 0.23 b | 4.17 | 0.56 |
| L* peel | 48.22 ± 2.12 a | 48.90 ± 2.21 a | 6.81 | 7.44 |
| a* peel | −13.52 ± 0.58 a | −14.60 ± 1.04 a | 10.6 | 3.34 |
| b* peel | 26.10 ± 0.92 a | 26.90 ± 0.67 a | 4.21 | 2.51 |
| C* | 30.62 ± 2.53 a | 29.42 ± 1.12 a | 7.02 | 4.43 |
| h* | 118.34 ± 1.71 a | 117.38 ± 1.45 a | 4.12 | 2.19 |
| L* pulp | 83.03 ± 3.28 a | 84.19 ± 2.93 a | 2.92 | 5.50 |
| a* pulp | 1.91 ± 0.15 a | −1.02 ± 0.09 a | 4.61 | 0.54 |
| b* pulp | 29.85 ± 1.03 a | 18.65 ± 0.58 b | 4.57 | 2.49 |
| C* | 29.68 ± 2.50 a | 18.68 ± 0.72 b | 4.03 | 2.84 |
| h* | 86.49 ± 1.20 b | 93.07 ± 0.80 a | 5.37 | 3.79 |

Means followed by the same lowercase letter on the same line do not differ from each other by Tukey's test ($p < 0.05$).

The fruits of the cultivar BRS Platina had a higher mass and pulp/peel ratio. Differences in this ratio indicate variations in the moisture content of the banana peel and pulp [36]. Pulp yield is an important quality characteristic for the industry. Varieties whose fruit has a high pulp yield can obtain higher yields after the final processing.

The fruits of cultivar BRS Platina had a greater firmness for the whole fruit and peel than those of FHIA 18, which may be due to the chemical composition of the peel. There was no difference in pulp firmness. Some studies indicate variations in the firmness of the whole fruit from 102.52 to 17.40 N, from 32.17 to 53.97 N for unripe pulp, and from 3.77 to 12.51 N for ripe pulp. The pulp to peel ratio of banana fruits increases with ripening, ranging from 1.18 to 2.26 (unripe) to 1.43 to 4.18 (ripe) [36–38].

The color analysis of the peel and pulp of the fruits showed that both cultivars had green peels, with no differences for all determined parameters. The pulp of the fruits of the cultivar BRS Platina had a creamier tone than those of FHIA 18.

Saturation is linked to the concentration of the coloring element and represents a quantitative attribute of intensity. The hue angle (h*) is considered a qualitative color attribute with the 0° angle indicating a red color, the 90° angle for yellow, the 180° angle for green, and the 270° angle for blue [37]. In the study by Aquino et al. [35], the peels of unripe fruits of different banana cultivars showed chroma (C*) variations from 8.65 to 28.21 with the lowest chroma for the cultivar Caru-Roxa and the highest for Maçã. The hue angle of the peels ranged from 66.42 to 122.17°, with the highest value for the Caju. The pulps had C* values ranging from 21.35 for the cultivar Marmelo to 35.62 for the Terrinha.

The hue angles ranged from 73.44° (Terrinha) to 93.29° (Marmelo). Thus, the results observed in our study are consistent with previous color data of unripe fruits of different banana cultivars.

The physicochemical characteristics of banana fruits are influenced by the cultivar, growing conditions, maturation stage, and post-harvest conditions. The pulp of the fruits of the cultivar BRS Platina differed mainly in terms of a lower total sugar content, higher lipid content, and higher acidity (Table 2). Our results agree with those observed in other studies with the same cultivars and in pulps of unripe fruit of different banana cultivars [35–41].

**Table 2.** Physicochemical characteristics of unripe banana fruits.

| Parameter | Cultivars | | CV (%) | MSD |
|---|---|---|---|---|
| | **BRS Platina** | **FHIA 18** | | |
| Moisture (%) | 66.28 ± 1.64 a | 67.24 ± 0.52 a | 1.94 | 2.92 |
| Total sugars (%) | 1.70 ± 0.04 b | 2.04 ± 0.03 a | 2.66 | 0.11 |
| Reducing sugars (%) | 1.04 ± 0.03 a | 1.10 ± 0.07 a | 3.42 | 0.33 |
| Lipids (%) | 0.45 ± 0.04 a | 0.11 ± 0.02 b | 15.75 | 0.10 |
| pH | 5.37 ± 0.09 a | 5.63 ± 0.13 a | 3.18 | 0.09 |
| Titratable acidity (mg/100 g) | 0.47 ± 0.11 a | 0.26 ± 0.09 b | 5.22 | 0.043 |
| Soluble solids (°Brix) | 3.85 ± 0.18 b | 4.32 ± 0.16 a | 3.63 | 0.99 |
| Ratio | 8.19 ± 0.13 b | 16.61 ± 0.10 a | 4.13 | 3.20 |

Means followed by the same lowercase letter on the same line do not differ from each other by Tukey's test ($p < 0.05$).

Moisture is an important quality parameter that affects the efficiency of the process, the amount of fat absorbed, and the texture (hardness and crispiness) of the fried products. The higher the moisture content, the greater the oil absorption and the less crispiness. In addition to being related to the fruit's state of maturity, the sugar content in the pulp is also an important parameter to assess its suitability as a raw material for the frying process. A high amount of reducing sugars induces negative changes (browning and off flavors) in processed products during frying [42,43].

*3.2. Characteristics of Banana Chips*

The amount of coating material adhered to the sample during immersion before frying varied according to the banana cultivar and the type of hydrocolloid (Figure 2). Greater weight gains were observed for the cultivar FHIA 18, which may be related to the lower firmness of pulp. Regardless of the cultivar, dipping banana slices in guar gum solution resulted in significantly greater weight gain (62.36% for BRS Platina and 73.56% for FHIA 18) than dipping in the other hydrocolloid solutions, which may be due to the higher viscosity of the guar gum solution resulting in better adhesion to the surface of the banana slices [19].

Guar gum consists mainly of high molecular weight galactomannan polysaccharides which are a linear chain polymer of β-D-mannopyranosyl units linked to (1→4) and with linkages (1→6) of α-D-galactopyranosyl as side chains. Mudgil et al. [44] point out that the most important characteristic of guar gum is the formation of highly viscous solutions, which may have favored weight gain (coating adherence).

Maltodextrins are starch hydrolysis products characterized by a dextrose equivalent value ranging from 0 to 20. This polysaccharide has the ability to form weak gels that are the result of interactions between helical amylose fractions and branched amylopectin molecules [45]. This characteristic of gels with lower viscosity may have interfered with the lower weight gains observed for this treatment (25%).

The moisture content in fried foods is extremely important to obtain a product with adequate mechanical properties, such as crispness. The moisture content decreased significantly after frying, with contents ranging from 3.56 to 9.22%, i.e., moisture losses between 86.3 and 94.62% (Figure 2). The water starts to evaporate as soon as the raw material

meets the oil, with a sudden drop due to the loss of surface moisture, and later reaches equilibrium [46].

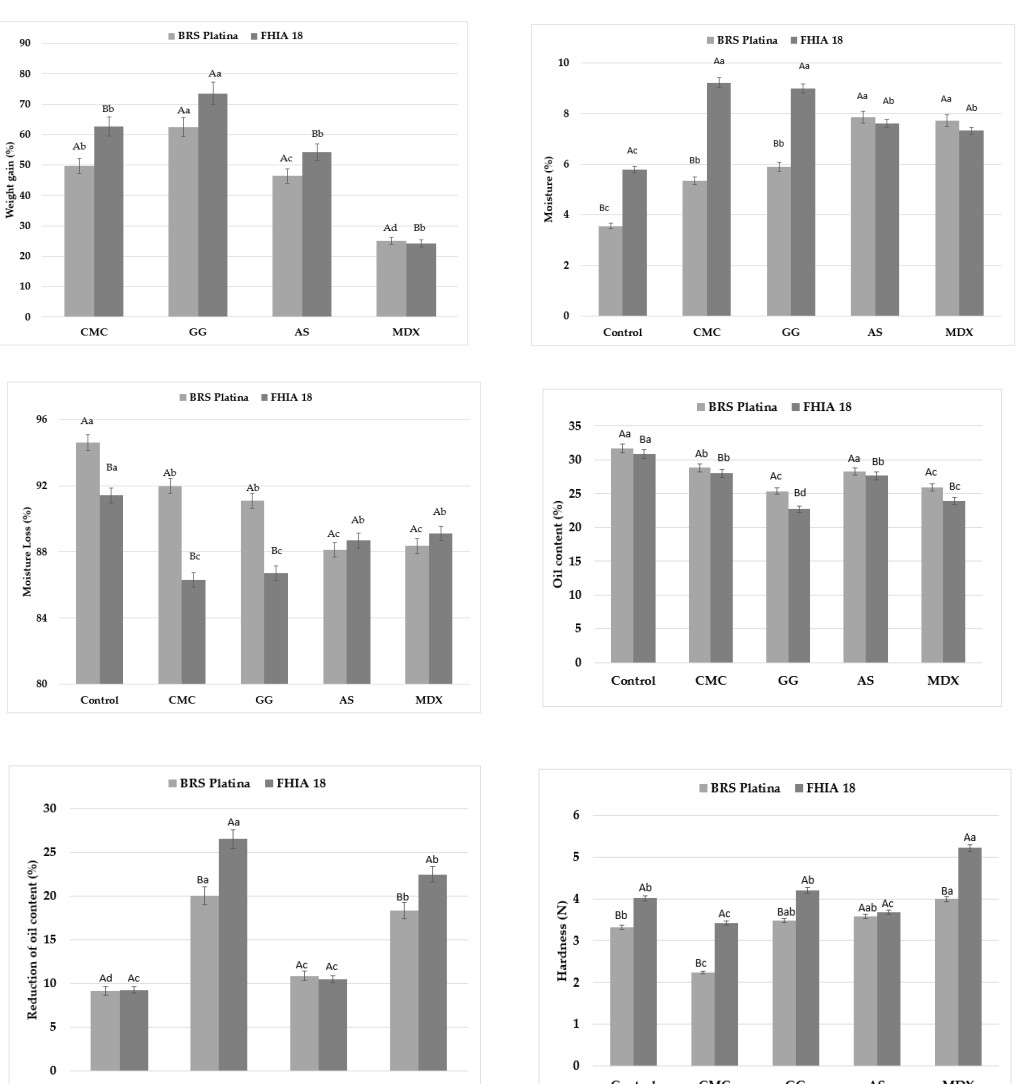

**Figure 2.** Weight gain, moisture, moisture loss, oil content, reduction in oil absorption, and hardness of banana chips from different pre-frying treatments. CMC = carboxymethylcellulose, GG = guar gum; SA = sodium alginate; MDX = maltodextrin. Means followed by the same uppercase letter within a treatment and the same lowercase letter between treatments do not differ according to Tukey's test ($p < 0.05$).

Banana chips obtained from different cultivars showed moisture values ranging from 1.03 to 5.46% [12]. The moisture content in the final product is the result of the raw material and processing characteristics, such as initial moisture, slice thickness and diameter, and frying time and temperature. The moisture retention ability of samples with different coatings was different from each other. There were significant differences in the moisture content between the banana chips coated with the different gums. The lowest moisture losses were observed for FHIA 18 chips treated with carboxylmethylcellulose and guar gum. The use of hydrocolloid coatings leads to moisture retention as the coating leads to an increase in the barrier property and prevents the creation of pores and cracks in the fried product [47]. In addition, the decrease in moisture loss from samples covered with hydrocolloids can be explained by thermogelation, which affects the water retention capacity of the samples during deep-frying [10].

Oil content is one of the most important quality indicators of fried foods, interfering with sensory and nutritional qualities and shelf life. The oil content in banana chips ranged from 22.70 to 31.72%, with the lowest values observed in fried products that were coated with hydrocolloid solutions (Figure 2). The banana chips from BRS Platina had higher oil contents, which may be related to the greater surface area in contact with the oil (larger diameter of the slices). The results observed by Pinzón et al. [12] for oil contents in banana chips ranged from 14.03 to 39.55% depending on the banana cultivar.

Our results showed reductions in oil absorption of between 9.14 and 26.54% (Figure 2), with better results in products coated with guar gum, similar to the results of Sumonsiri et al. [13] who studied the production of chips of banana coated with guar gum and observed an oil content of 43.25% in the control chips with a reduction of 21.18% using immersion in a solution with 1% guar gum. Characteristics such as high solubility and low viscosity of the maltodextrin solution may have contributed to keeping the oil evenly distributed, causing a high fat reduction in the chips (Figure 2).

Singthong and Thongkaew [10] studied pre-treatments in the process of producing fried banana chips and also observed a low reduction in oil content in products bleached with 0.5 g $CaCl_2$/100 mL of water and immersed in a 1% solution ($w/v$) of sodium alginate; they reported that the best result in the reduction of oil absorption and sensorial attributes was reached in the treatment with 0.5 g of $CaCl_2$/100 mL and 1% pectin.

Variations in moisture and oil contents and in the reduction in oil absorption in fried plantain chips as a function of the type of hydrocolloid and time of immersion in the solution were observed by Martinez et al. [48]. The authors reported 1.25% moisture in control chips, ranging from 2.50 to 3.12 for those coated with carboxymethylcellulose, and from 2.53 to 3.61% for guar gum. Oil contents ranged from 21.18 to 24.01 for CMC and 17.5 to 22.91% for GG, with reductions in oil absorption from 21.79 to 31% for CMC and 25.37 to 43.01 for guar gum. Furthermore, pre-treatments with hydrocolloids improve sensory quality attributes such as crispness, final flavor, and general acceptability, which relate to the perception of oil content in the final product [49].

The breaking force or hardness is an indicator of the extent of crispness. A lower hardness value corresponds with a higher crispness. The chips produced from the FHIA 18 cultivar had a lower crispness than those obtained from the cultivar BRS Platina, with the best result observed for the chips covered with CMC for this cultivar. The chips from FHIA 18 had a greater weight gain, and greater moisture and hardness. That is, a thicker layer of hydrocolloid may have formed and acted as a barrier to the movement of water from the inside to the outside of the banana slices during frying, and thereby requiring greater force to break the sample [14,16].

Paramasivam et al. [49] observed in their study with banana chips that hardness increased with increasing hydrocolloid concentration, suggesting that this result may be due to the inherent property of hydrocolloids that provides a higher mechanical strength to banana chip tissues, which in turn, resulted in improved structural integrity and therefore less crispiness.

The overall appearance of a fried product is of great importance for consumer acceptance, and color plays a key role in appearance. The golden brown color of fried chips is a result of deep fat frying, resulting in lipid oxidation as well as carbohydrate–protein-driven caramelization and Maillard reactions [50].

The color of the banana chips was influenced by the cultivar and the hydrocolloid treatments (Figure 3). The BRS Platina chips had higher L*, a*, b* and chroma values. The hue angle values of all chips were above 70°, showing the transition from red to yellow and indicating the development of a golden brown color, which is highly appreciated by consumers of banana chips [51]. The application of the coatings increased the L* value of the banana chips compared to the control treatment and decreased the browning index. The FHIA 18 chips showed a less intense brown color than the BRS Platina chips. The control chips had higher browning indices than the coated chips, with a lower BI for the maltodextrin-coated chips (Figure 3).

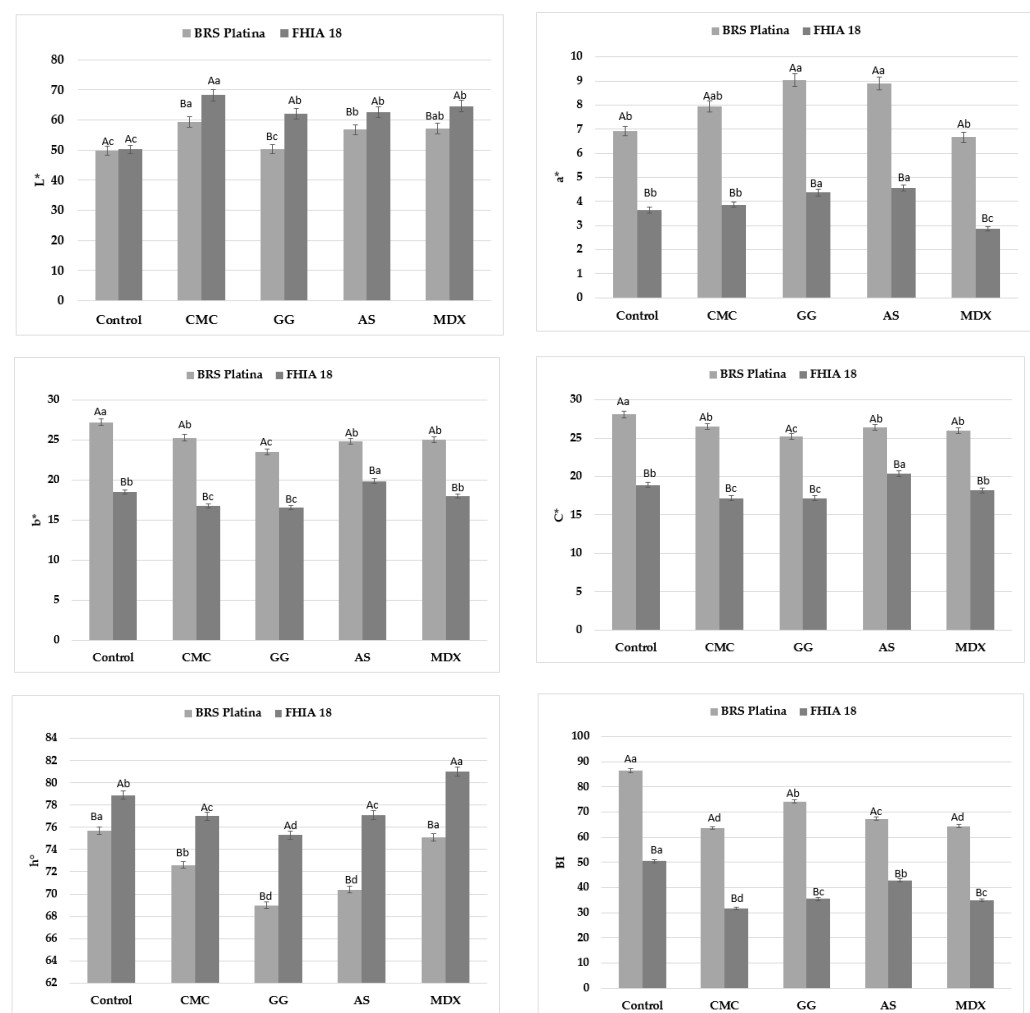

**Figure 3.** Color of banana chips from different pre-frying treatments. CMC = carboxymethylcellulose, GG = guar gum; SA = sodium alginate; MDX = maltodextrin. Means followed by the same uppercase letter within a treatment and the same lowercase letter between treatments do not differ according to Tukey's test ($p < 0.05$).

The lower browning of the banana chips covered with the hydrocolloids is due to the fact that the coating on the sample surfaces may have acted as a barrier against the transfer of heat from the oil to the samples, reducing dehydration due to the hydrocolloids' ability to bind to water, which may have interfered with the Maillard reaction [14]. Al-Asmar et al. [52] demonstrated a reduction in the Maillard reaction in coated fried products and concluded it was due to increased water retention during frying.

## 4. Conclusions

Product diversification through processing is important to make banana production more sustainable for small growers. Furthermore, the processing sector and consumers demand high technological and nutritional quality for fried products. Thus, our findings showed that the production of fried chips from BRS Platina and FHIA 18 cultivars with lower oil contents is interesting. Cultivar FHIA 18, despite producing slices with a smaller diameter, produced fried chips with less oil and a lighter color. The chips from the BRS Platina cultivar had a lower moisture content and greater crispness. Pre-treatment with guar gum reduced the oil content by more than 20%, followed by maltodextrin, which contributes to the healthiness of the product.

**Author Contributions:** Conceptualization, M.L., S.L. and A.M.F.; methodology, validation, formal analysis, and investigation, J.S.P.d.S., P.R.R.d.J. and L.F.d.O.; data curation, writing—original draft preparation, and writing—review and editing, J.S.P.d.S. and M.L.; supervision, project administration, and funding acquisition, M.L. All authors have read and agreed to the published version of the manuscript.

**Funding:** The authors acknowledge the financial support of the National Council for Scientific and Technological Development (CNPq) (Grant numbers 302848/2021-5 and 302611/2021-5).

**Institutional Review Board Statement:** Not applicable.

**Informed Consent Statement:** Not applicable.

**Data Availability Statement:** Data are contained within the article.

**Conflicts of Interest:** The authors declare no conflict of interest.

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
