# Peer review of "Hydrocolloid Coatings as a Pre-Frying Treatment in the Production of Low-Fat Banana Chips"

_horticulturae, doi:10.3390/horticulturae9101139_

Round 1
Reviewer 1 Report (Previous Reviewer 4)
The authors have addressed my previous comments and revisions made by the authors make this paper clearer. The paper can be accepted.
Author Response
Dear reviewer,
We appreciate your important review of our manuscript.
Thank you very much for taking the time to help us.
Yours sincerely,
Authors
Reviewer 2 Report (Previous Reviewer 1)
The manuscript has been explained and revised in accordance with the reviewer's comments.
Author Response
Dear reviewer,
We appreciate your important review of our manuscript.
Thank you very much for taking the time to help us.
Yours sincerely,
Authors
Reviewer 3 Report (Previous Reviewer 2)
Accept.
Author Response
Dear reviewer,
We appreciate your review of our manuscript.
Yours sincerely,
Authors
Reviewer 4 Report (Previous Reviewer 3)
What is the relation of characteristics of unripe fruits of banana cultivars with the frying process? Clearly mention what was the need of this, as the study is related to coatings and frying.
All the data is presented in the form of tables. Why? As a matter of fact, figures are more illustrative than the tables.
Delete the sentence in the abstract: “Despite the large volume of banana fruit produced, there are high rates of post-harvest losses and processing is one way of reducing production waste.”
Change “banana fruits cultivars” to “banana fruit cultivars”
Better to revise the last sentence in the abstract, specifically remove the statement “Our findings are useful for expanding germplasm diversification in orchards,”
Combine first two paragraphs of Introduction. Throughout the documents, there are some other paragraphs that consist of few lines. Combine similar paragraphs to have an appropriate length of each paragraph.
Remove repetition regarding production of bananas, lines 31 and 43.
Line 43: The statement “cheap source of energy” might not be true for all countries.
Line 59: Better to say “Frying is a thermal process”
Provide suitable reference for color measurement, line 192.
Figure 1 is referred on line 209. However, it should be referred earlier in section 2.4.
Moderate editing of English language required.
Author Response
We thank you for your careful review of our manuscript and your considerations aimed at improving the work.
All your corrections to the text have been accepted.
The reference for the color analysis has been added.
Figure 1 has been correctly positioned in the text.
The importance of analyzing the physical and chemical characteristics of the fruit has been discussed in the text.
Tables of banana chip results have been replaced by graphs.
We hope we have responded to your suggestions, however, if you consider that additional changes to our manuscript are necessary, we are willing to make them.
Yours sincerely,
Authors
Round 2
Reviewer 4 Report (Previous Reviewer 3)
The authors have considered majority of the comments, so the article can be accepted.
Minor editing of English language required
This manuscript is a resubmission of an earlier submission. The following is a list of the peer review reports and author responses from that submission.
Round 1
Reviewer 1 Report
In this manuscript, the author carefully analyzed the waste in the banana industry caused by its high perishability and immature production technology. To reduce waste, the production of fried banana chips has become an alternative solution to minimize production losses. However, oil absorption during the frying process has become a key factor for consumers in choosing banana chips. Therefore, the author analyzed the physicochemical characteristics of banana varieties to serve as the raw material for fried banana chips. Then, four types of hydrocolloids were used as pretreatment in the frying process, and the quality of the fried banana chips was analyzed based on physicochemical indicators such as oil content, texture, color, and moisture content. The analysis of banana varieties in the manuscript provides new ideas for banana growers and helps expand the genetic diversity of orchards. Hydrocolloids' coating treatment on banana chips effectively reduces the oil content of fried banana chips and promotes the development of the banana market. However, the manuscript also has the following issues:
Question 1: The description in line 23 "Hydrocolloids promoted reductions in oil absorption from 9.14 to 26.54%...." is incorrect and should be changed to "from 26.54% to 9.14%."
Question 2: The format for the percentage changes in the oil content of banana chips should be "from 1 percent to 2 percent," it appears multiple times in the manuscript. Please change one by one.
Question 3: The format of a space between paragraphs in lines 45-47 is incorrect and should be corrected.
Question 4: Since bananas come in different sizes and have an uneven surface, measuring the hardness at only one point may result in significantly different values. The manuscript should provide an explanation or additional experiments on this matter. In addition, the crispness of banana chips is also an important factor affecting consumer taste and should be included in relevant experiments.
Question 5: In line 172, “moisture of raw slices” should be clarified to indicate whether it refers to “slices treated with hydrocolloids” or “slices without pretreatment.” Question 6: All data in Table 1 and Table 2 should be centered, and the format of “CV (%)” and “MSD” should be adjusted accordingly.
Question 7: The manuscript lacks standard deviation data, and it should be supplemented.
Question 8: Acrylamide could be produced during the frying process, and it may affect consumers’ choices of fried banana chips. The manuscript should consider including it as an important factor.
Question 9: The manuscript mentions that consumer sensory evaluation is essential, but it only refers to hardness and color. Other sensory evaluation indexes, such as “flavor” and “appearance” should also be considered.
The English expression of the manuscript can be further improved.
Author Response
We are grateful for your careful review of our manuscript and for your considerations aimed at improving the work.
We made specific corrections to the text and formatting (Questions 1 to 3).
The firmness of fresh fruits was determined at three points on the fruits and the data was added to Table 1 (Question 4).
The relationship between hardness and crispness of banana chips and the discussion of these parameters has been improved in the text (Question 4).
The calculation of moisture loss was better explained (Question 5).
Tables 1 and 2 were adjusted and standard deviations were inserted (Questions 6 and 7)
Unfortunately, the acrylamide content was not determined. However, we included information about reduced formation of acrylamide in fried banana chips obtained from green fruits, also about the reduction of acrylamide formation in banana chips with the use of hydrocolloids (Question 8).
Due to the restrictions imposed by our University on carrying out sensory tests due to Covid, we were unable to carry out this important evaluation, but we have included in the text information on the effects of pre-treatments with hydrocoloids on banana chips for sensory acceptance (Question 9 ).
We appreciate your important contribution and we can make any other corrections you deem appropriate.
Yours sincerely,
Authors
Reviewer 2 Report
Ref. No.: horticulturae-2629043
Manuscript: Hydrocolloid coatings as a pre-frying treatment in the production of low-fat banana chips
Journal: Horticulturae (ISSN 2311-7524)
The current study belongs to the field of food processing, but the experimental design is too simple, only a simple analysis of the basic physical and chemical properties of two kinds of bananas, and then a comparative study of fried banana slices treated with different hydrocolloids. However, the processing process did not analyze the parameters of each factor, such as processing time, addition amount, etc., which could not reflect the real difference. On the whole, the current research is not innovative enough, the experimental design lacks integrity, can not solve the fundamental problem.
Author Response
We appreciate your careful review of our manuscript and your considerations aimed at improving the work.
Our study evaluates two cultivars with good agronomic performance but which have had little acceptance by growers due to their lower commercial value due to their easy fruit drop.
In this first study, it was important for us to verify the efficiency of various hydrocolloids in reducing the oil content and other quality attributes of fried chips, so that in a second experiment we could, through a more complex experimental design, achieve optimization of the processing conditions.
Our results are significant because they show the relevance of the intrinsic characteristics of each cultivar as determining factors for indication as raw materials for fried chips, and also the effectiveness of all hydrocolloids in reducing the oil content in the final product, contributing to the definition of hypotheses for future research.
We are aware of the importance of continuing the research and can make any corrections to our study that you point out as important for improving its quality.
Yours sincerely,
Authors
Reviewer 3 Report
Since the article primarily focuses on processing, it is advisable not to submit this article for publication in Horticulturae.
Reviewer 4 Report
This article evaluates the physicochemical characteristics of banana fruit varieties and the effects of hydrocolloids coating on the quality attributes of banana chips. The research results obtained are quite interesting, but the article still needs some improvement.
1. Only two banana varieties were selected as research materials in the article. Please explain whether these two varieties of bananas are representative in cultivation and food processing.
2 Due to the presence of polysaccharides, including starch and cellulose, as well as the protein content in banana pulp, different banana varieties may differ in color and properties after frying. The author did not analyze the protein content of banana pulp. It is recommended to make some inferences based on the research results of relevant references in discussion.
3 Banana pulp is prone to browning under the action of polyphenol oxidase. Is the degree of browning between these two varieties related to differences in polyphenol oxidase activity? Is coating helpful in inhibiting oxidative browning of fruit pulp? It is also recommended to supplement relevant content in discussion.